# Assessment of time-related deficits in older adults: a scoping review protocol

Sebestina Anita Dsouza ,[1] Meena Ramachandran ,[2] Yuko Nishiura,[3] Bhumika Tumkur Venkatesh ,[4] Lena Dahlberg [5,6]

¹Department of Occupational Therapy, Centre for Studies on Healthy Aging, Manipal College of Health Professions, Manipal Academy of Higher Education, Manipal, Karnataka, India
²Faculty of Health Sciences, McMaster University, Hamilton, Ontario, Canada
³Department of Assistive Technology, National Rehabilitation Center for Persons with Disabilities Research Institute, Tokorozawa, Saitama, Japan
⁴Public Health Evidence South Asia, Manipal Academy of Higher Education, Manipal, Karnataka, India
⁵School of Health and Welfare, Dalarna University, Falun, Dalarna, Sweden
⁶Aging Research Center, Karolinska Institutet and Stockholm University, Solna, Sweden

**Correspondence to**
Dr Lena Dahlberg; ldh@du.se

## ABSTRACT

**Introduction** People with cognitive impairments often have difficulties in managing their time for daily activities. In older adults with cognitive impairments such as dementia and stroke, these may present as disorientation, poor time awareness, time perception, daily time management and so on. Time-related deficits and associated behaviours impede independent living and add considerably to caregiver strain. Several interventions are being investigated to help people with cognitive impairments orient and navigate time and do their daily activities. The provision of interventions requires the use of sound assessment tools. However, it is not clear how time-related concepts are specifically evaluated in practice, what are the available assessments and how these assessments should be selected.

**Method and analysis** This protocol follows the Joanna Briggs Institute Reviewer's Manual (2020) for scoping reviews and is registered with the Open Science Framework (https://osf.io/4ptgy/). We will include the following databases: PubMed, CINAHL, Scopus, Web of Science and PsycINFO. Two reviewers will independently screen eligible studies for inclusion against the selection criteria and then review the full-text of the selected studies. We will extract the bibliographic data, study design and setting, and details of assessments used in the studies to evaluate time-related concepts including format, mode and duration of administration, psychometric properties and so on. The identified assessments will be mapped with regard to time-related concepts being evaluated and described using narrative synthesis.

**Ethics and dissemination** As secondary data analysis, ethics approval is not required for this scoping review. We plan to disseminate the results through peer-reviewed journals and conferences targeting health professionals working with older adults.

## Strengths and limitations of this study

► The review will be comprehensive as we will be screening all studies published until the completion of our search.
► The protocol involves a rigorous methodological framework recommended by Joanna Briggs Institute.
► We may miss studies published in languages other than English and not available in databases included in this protocol.

## INTRODUCTION

The ageing population is increasing worldwide and is expected to double by 2050.[1 2] While there are various health problems that affect older adults, neurological disorders are the most prevalent and debilitating. The disability-adjusted life years due to these disorders are significant and are expected to increase with the ageing population.[3] Cognitive impairments are a common clinical feature of several neurological disorders.

Among neurological disorders, dementia in older adults is a salient concern.[4] Dementia is a syndrome that has been described as a global epidemic, as the incidence and prevalence of dementia are increasing exponentially, especially in low-income and middle-income countries.[5 6] Besides dementia, cognitive impairments are common in an array of neurological conditions such as stroke, epilepsy, traumatic brain injury, Parkinson's disease and so on, and can range from subclinical to severe impairment.[7 8] Cognitive impairments, even when mild, are known to impact quality of life[7 9] and participation,[10] and contribute significantly to caregiver burden.[11]

Cognitive impairments include deficits in attention, memory, executive functions, learning capacity and so on, that impacts activity participation.[10] Thus, participation in daily activities also requires cognitive functions that are specifically related to time. These time-related cognitive functions help in planning and coordinating various daily occupations into patterns of time forming routine.[12] Several terms have been used to describe time-related abilities and cognitive functions. For convenience, in the present study, these are collectively referred to as time-related concepts and include time-related cognitive functions and daily time management abilities.

The International Classification of Functioning, Disability and Health (ICF) describes several time-related cognitive functions under

the domain of body functions.[13] These include orientation to time (b1140) and experience of time (b1802) and also time management (b1642), the latter as a component of executive functions (b164). To facilitate better understanding, Janeslätt has organised the various time-related cognitive functions in a three-level hierarchy within a single construct of 'time processing ability' (TPA).[14 15] Within this construct, time perception refers to the experience of time and is at the lowest level of this hierarchy. It entails the ability to know how time passes and the duration of different activities. Time orientation is at the next level of TPA and involves awareness of time such as the time of the day, date, day of the week, month and year. Finally, time management is at the highest level of TPA and refers to the ability to plan and order daily activities chronologically and to know how much time to allocate for these activities.[14 16] Time management is a cognitive function under TPA and is distinct from the concept of daily time management.

Time is an integral and salient component in the performance of daily activities. This is well described in the activity and participation domain of the ICF which has, in all, seven components related to the use of time in daily activities.[13] One example is 'carrying out daily routine' (d230) under general tasks and demands (d2). This includes managing daily routine (d2301), completing daily routine (d2302), managing one's own activity level (d2303), adapting to changes in daily routine (2304), managing one's time (d2305), carrying out specified (d2308) and unspecified (d2309) daily routines. This shows how time is embedded in this complex component of carrying out one's daily routine and cuts across other components in the domain of activity and participation. Managing one's time (d2305) is described as 'managing the time required to complete usual or specific activities, actions and behaviours appropriately in the required sequence and within the time allotted', and entails managing one's time and adapting to time demands.[17] It is also known as daily time management.[18]

People with cognitive impairments are known to have difficulties in managing their time for their daily activities.[19–21] Difficulties in time-related cognitive functions could present in different ways in different populations of older adults. Persons with dementia are known to have disorientation to the time, day and date, require more time for daily activities, difficulty in keeping track of past events, planning for future events, difficulty knowing, remembering and planning 'when' and difficulty in judging the duration of different activities and situations.[22] Haj and Kapogiannis have described time distortions in dementia due to deficits in memory and time perception.[23] Difficulties in time estimation have been attributed to deficits in central executive function[24] and distortions in different components of the internal clock.[25] Requena-Komuro and colleagues[26] have reported altered time awareness that contributes to abnormal behaviours in dementia syndromes such as clockwatching, reliving past events and temporal inflexibility and the Godot

syndrome, that is, anxiety symptoms regarding upcoming events by repeatedly asking questions. Difficulties with time estimation have been reported in individuals with mild cognitive impairments[25 27] and also in older adults without cognitive impairments.[28]

Time-related deficits and associated behaviours affect the ability to do daily activities independently and add considerably to caregiver strain and distress. The ability to manage one's time effectively is an essential factor influencing a person's self-efficacy.[29 30] With increasing understanding of the importance of time-related abilities on performance of daily activities, several interventions are being investigated to help people with cognitive impairments orient and navigate time and do their daily activities. These include reality orientation,[31] high-tech assistive devices such as time aids, automatic calendars,[22 32 33] electronic devices,[34] mobile-based applications[35] and low-tech assistive devices such as paper-based calendars,[36] routine organisers,[37] and training to improve organisational skills and habits.[38 39]

The provision of interventions require the use of sound assessments to screen, evaluate or identify the nature and severity of time-related cognitive deficits and their impact on performance of daily activities. They are also essential to monitor and document changes with intervention or disease progression. Assessments have also been used by occupational therapists to design interventions for children and youth having difficulties with time-related concepts.[40 41] However, it is not clear how time-related cognitive functions and daily time management of older adults are evaluated in practice, what are the available assessments and how these assessments should be selected.

A preliminary search for existing reviews on this topic was done in the following databases: Joanna Briggs Institute (JBI) database of Systematic Reviews and Implementation Reports, Cochrane Database of Systematic Reviews, Cumulative Index to Nursing and Allied Health Literature (CINAHL), PROSPERO and PubMed. We did not find any review addressing assessment of time-related concepts in older adults. The proposed scoping review intends to address this lacuna by systematically mapping research done in this area.

### Research question
What are the available tools to assess time-related cognitive functions and daily time management in older adults?

### Aim
The aim of this proposed scoping review is to investigate assessments tools for time-related cognitive functions and daily time management in older adults with and without cognitive impairments.

### Objectives
► To map the assessments tools available for older adults with respect to time-related concepts.
► To describe the characteristics of available assessment tools including, purpose, type and format of the

test, study populations, psychometric properties and feasibility.

## METHODS

The protocol was drafted using the JBI Reviewer's Manual.[42] The protocol is registered with the Open Science Framework (https://osf.io/4ptgy/). The review will be undertaken between August 2020 and March 2022 and will involve the following steps:

### Identification of relevant studies

The selection criteria and search strategy to identify relevant studies are as follows:

► Selection criteria:
- Articles with older adults as participants, that is, people aged 60 years or older OR groups with an average age of 60 years or older,[5 43] with and without cognitive impairments will be included. Articles that have older participants along with other age groups will also be included.
- Primary research articles (quantitative and mixed method studies) that discuss assessment tools for one or more time-related concepts described in the introduction, either as a separate/specific test or as a component of another test (such as a cognitive assessment), if results of that component are presented separately. This includes:
  - Pre–post, quasi-experimental and experimental (randomised and non-randomised controlled trials) designs evaluating the effectiveness of interventions targeting time-related concepts (if the outcome measures used include assessments of time-related concepts) and also observational studies investigating one or more time-related concepts. Articles with experimental laboratory-based designs will be excluded.
  - Validation studies, that is, studies developing or validating an instrument or assessment tool that assesses one or more time-related concepts.
- Peer-reviewed articles in English to date will be included. The language restriction to English is based on convenience and feasibility of the authors involved in the study group.
► Search strategy: The key words will be identified and search strategy will be constructed with involvement of all the authors. The search will be conducted in the following databases: PubMed, CINAHL, Scopus, Web of Science and PsycINFO. First an initial limited search of at least two databases will be done such as PubMed and CINAHL. This will be followed by analysis of the text words contained in the title and abstract of the retrieved publications and the index terms used to describe the articles. A second search using all identified keywords and index terms will be undertaken across all included databases. The reference list of included publications will be searched for additional sources. The bibliographic information of included publications will be combined and stored

| Table 1 | Keywords for search |
|---|---|
| **PICO** | **Keywords** |
| Population: Older adults with and without cognitive impairments | elderly, older, older age, aged, older adults, elderly persons, old, seniors, aging adults, older persons, geriatric, executive dysfunction, mild cognitive impairment, dementia, delirium, cognitive impairment, cognitive dysfunction, cognitive limitation, Alzheimer's, senile, psychiatric disorders, neurological disorders, stroke, Parkinson's. |
| Intervention: Assessment tools | assessment tool, screening tool, evaluation tool, diagnostic, psychometric properties, psychometrics, validity, reliability, test, prediction, predictive value, questionnaire, instrument. |
| Comparison: Not applicable | – |
| Outcome: Time-related concepts | time, time sense, time perception, time orientation, time awareness, time processing abilities, time management/managing one's time, routines, schedule, orientation, disorientation, time planning, executive functions, cognitive functions, cognition, cognitive abilities, judging time, time judgement, time organization, organizing time, task duration, activity duration, temporal, temporality, calendar. |

using EndNote. The PICO (Population, Intervention, Comparison, Outcome) and keywords for the search are summarised in table 1. See online supplemental material for an example. The search terms will be refined further if required. A combination of descriptors (eg, MeSH (Medical Subject Headings) terms) and keywords will be used with appropriate Boolean operators. The search strategy will be tailored to each database. We plan to conduct the search between September and October 2020. An updated search will be carried out later during the review period.

### Selection of articles (relevance)

Two reviewers will independently review the title and abstract of the included identified publications against the selection criteria. They will be marked as 'potentially relevant' or 'potentially irrelevant'. Discrepancies, if any, will be resolved by discussion and consensus between the two reviewers. If not resolved, a third reviewer will be consulted. This will be pilot tested using the framework proposed by JBI.[42] Full-text publications will be retrieved for studies regarded as 'potentially relevant', and will be reviewed independently and in duplicate for inclusion against the selection criteria. All reviewers will use a pilot-tested screening form developed for this review.

### Data extraction

Two reviewers will independently review the included articles using a data extraction form. The data extraction form will be adapted from JBI.[42] It will be piloted for

approximately five articles. Modifications, if any, will be made and the remaining articles will be reviewed with the finalised data extraction form. The details to be extracted would include:

1. Bibliometric data: title of the study, journal, year of publication, place (country) where the study was conducted.
2. Study details: aim/purpose/research question, study design, study setting (institution, community, inpatient, etc), study population (older adults with cognitive impairment, without cognitive impairment, etc), mean age of participants, sample size, recruitment strategy and so on.
3. Details of time-related assessments used in the study: time-related concepts being assessed, purpose/type of assessment (screening, diagnostic), language, format (computer/paper–pencil), mode of administration (self-report/therapist-administered; questionnaire/interview/observational), cost of using the assessment tool (paid/free), time for the test, psychometric properties and so on.

### Collating, summarising and reporting the results

An overview of the included publications will be provided in a tabular format to summarise the various aspects such as study type, aim, study design, settings, participants and time-related cognitive assessments used. As it is a scoping review, quality assessment of the identified articles will not be undertaken. The identified assessments will be mapped with regard to time-related concepts being assessed. A narrative synthesis will be carried out for test characteristics and to summarise the findings. Reporting will be done using the Preferred Reporting Items for Systematic reviews and Meta-Analyses extension for Scoping Reviews.[44]

### DISCUSSION

The present protocol outlines a scoping review of peer-reviewed articles on assessments available for time-related concepts in older adults and to our knowledge, this is the first of its kind. This review will advance knowledge about assessing time-related concepts in older adults. This area is gaining importance due to its impact on function and participation of older adults and their caregivers. The review will inform health professionals such as occupational therapists on the selection of appropriate instruments to evaluate different time-related concepts in older adults and guide clinical reasoning in planning interventions to support daily time management in older adults. This includes the provision of time management products. These products are in the top 50 priority Assistive Product List identified by WHO.[45] Older adults are expected to be major consumers of assistive products. In line with the mandate of United Nations Convention on the Rights of Persons with Disabilities, member states are required to enable affordable access to these devices, making this a priority area in rehabilitation across countries. Global

initiatives in this area would require the use of appropriate, valid and reliable assessments as well as appropriate assistive devices that are less likely to be abandoned by older adults in need of support.[46 47] Thus, provision of suitable products and adjunct interventions necessitates detailed evaluation of end users needs as suggested by the first global research, innovation and education on assistive technology summit.[48] These assessments would also help to monitor and determine the effectiveness of time management products and objectively study the impact of these interventions on other variables such as caregiver burden, quality of life, well-being, occupational participation and so on. Finally, the review will identify gaps to be addressed in future research. If sufficient literature is identified, a future systematic review or meta-analysis may be warranted.

### ETHICS AND DISSEMINATION

As a secondary analysis, ethics approval is not required for this scoping review. We plan to disseminate the results through peer-reviewed journals and conferences targeting health professionals working with older adults. Amendments to the protocol, if any, will be stated in any future publication based on the review presented in this protocol.

**Acknowledgements** The authors would like to thank Dr Gunnel Janeslätt, Researcher, Center for Clinical Research, Dalarna University, Dalarna, and associated to Centre for Public Health and Caring Sciences, Disability and Habilitation, Uppsala University, Uppsala, Sweden, for her valuable guidance and support to this scoping review protocol.

**Contributors** All authors have made substantive intellectual contributions to the development of this protocol. SAD conceived the idea of this research, followed by discussions with the other authors that contributed to finalising the research idea. All authors worked on the methodology of the scoping review. SAD drafted the protocol with inputs from MR, YN and BTV, which was further reviewed and revised by LD.

**Funding** This scoping review is part of an ongoing international research collaboration 'Managing Time in Dementia' funded by the Indian Council of Medical Research (54/1/GER/Indo-Sweden/17-NCD-II), India; FORTE, (FORTE 2017-00029) Sweden; and Japan Society for the Promotion of Science, Grant-in-Aid for Young Scientists (B) (16K16458), Japan.

**Competing interests** None declared.

**Patient consent for publication** Not required.

**Provenance and peer review** Not commissioned; externally peer reviewed.

**ORCID iDs**
Sebestina Anita Dsouza http://orcid.org/0000-0001-9193-9600
Meena Ramachandran http://orcid.org/0000-0003-4670-5375

Bhumika Tumkur Venkatesh http://orcid.org/0000-0002-3338-6478
Lena Dahlberg http://orcid.org/0000-0002-7685-3216

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
