## [Reviewer comments · BMJ Open]

ARTICLE DETAILS

TITLE (PROVISIONAL)	Assessment of time-related deficits in older adults: A scoping review protocol
AUTHORS	Dsouza, Sebestina; Ramachandran, Meena; Nishiura, Yuko; VENKATESH, BHUMIKA; Dahlberg, Lena

VERSION 1 – REVIEW

REVIEWER	Rodriguez, Francisca Technische Universitat Kaiserslautern
REVIEW RETURNED	12-May-2021

GENERAL COMMENTS	The authors describe a challenging but important project that aims at summarizing instruments for assessing orientation in time in older adults. I have only a few minor comments: 1. I would not agree to the use of the term "managing time". Time is a condition that is stable and therefore it cannot be managed. It is always the person who is actively adjusting his or her behavior to the temporal context he or she is in. I suggest that the authors adjust the wording in the manuscript accordingly.2. There is a conceptual challenge when discussion orientation in time and daily routines at the same time. Routines are often triggered by stimuli and carried out as habits, almost automatically. Orientation in time on the other hand is an active integration of the self in the external context. During the course of dementia the ability to make sense of numbers and quantities get lost and so does the understanding of time points and duration throughout the day. Accordingly, orientation is difficult because the concepts in the brain cease to exist. Habits (in daily routines), however, are often still carried out for a very long time during the course of dementia. The authors might want to wish to clarify what they mean in the introduction.3. In the methods, I want to motivate the authors to work with the PICO criteria even when it is not straight-forward in this case. "P" and "C" is very clear. "I" would refer to the assessment (what types of assessment do you consider) and "O" would refer to the concept of time that is being measured. You could specify this under point 1. in the methods section. Especially because I think the authors will have to provide more details of the concept of time that the instruments measure in the data extraction (point 3.)4. Regarding point 3. Data extraction, I suggest the authors to also extract the recruitment strategy, country, and the mode of the instrument (questionnaire, interview, observational ...).5. In the discussion, what is the bigger goal? What exactly do you want to do with the tool in the future and how is it going to improve patients' lifes?
--

REVIEWER	Ayyala, Deepak Nag Augusta University, Population Health Sciences
REVIEW RETURNED	17-May-2021

GENERAL COMMENTS	The aim of this protocol is to investigate various available on how time-related concepts are evaluated in understanding the time perception of older adults with cognitive impairments. The authors aim to perform a meta-analysis of studies published by screening various databases. The purpose of the study and the approach are very clearly presented. However, there are a few minor comments that can be addressed to improve the protocol. Major comments:  1. For age: the selection criterion is not very stringent. The authors should define what "older adults" mean, (e.g. age greater than 50) use that for their search. Using average age might not be ideal and should be removed. If not possible, they should consider the median to be more robust to outliers. 2. There should be a minimum sample size requirement for an article to be included in the analysis. Experimental protocols with a small number of replicates might introduce more variability into the analysis. 3. A detailed statistical analysis plan for doing the meta-analysis is not provided. The authors could provide some details on the overall plan of how to do the analysis before reporting the results. Minor comment:  1. On Page 10, line 50: the range of dates for the search is mentioned as September-October 2020. Please correct it to a future date if the data has not been collected yet. Otherwise, please provide preliminary data. 2. All the references are mentioned after a comma or a period, instead of before. Please fix them.
--

VERSION 1 – AUTHOR RESPONSE

Reviewer: 1

Dr. Francisca Rodriguez, Technische Universitat Kaiserslautern

Comments to the Author:

The authors describe a challenging but important project that aims at summarizing instruments for assessing orientation in time in older adults. I have only a few minor comments:

1. I would not agree to the use of the term "managing time". Time is a condition that is stable and therefore it cannot be managed. It is always the person who is actively adjusting his or her behavior to the temporal context he or she is in. I suggest that the authors adjust the wording in the manuscript accordingly.

Response: The concept and term 'managing time' is widely used in research and drawn from ICF. In a particular sentence on p. 4, the cognitive function of time management is meant in contrast to the person's ability to manage one's time for daily activities. To clarify this sentence and paragraph, we have omitted these words as the concepts are further described based on ICF in the following paragraph.

2. There is a conceptual challenge when discussion orientation in time and daily routines at the same time. Routines are often triggered by stimuli and carried out as habits, almost automatically. Orientation in time on the other hand is an active integration of the self in the external context. During the course of dementia, the ability to make sense of numbers and quantities get lost and so does the understanding of time points and duration throughout the day. Accordingly, orientation is difficult because the concepts in the brain cease to exist. Habits (in daily routines), however, are often still carried out for a very long time during the course of dementia. The authors might want to wish to clarify what they mean in the introduction.

Response: The concepts related to time are taken from ICF and work by Janeslätt (see references in manuscript). Following the definitions in ICF and provided by Janeslätt, orientation to time is a body function that comes under cognitive functions and thus in ICF a narrower concept than "an active integration of the self in the external context". Still, orientation to time is required for a person to carry out his or her routines, alter or modify routines, etc. Routines are behaviors and are classified under activity and participation of ICF as described in our introduction 'carrying out daily routine (d230)' under general tasks and demands. This includes managing daily routine (d2301), completing daily routine (d2302), managing one's own activity level (d2303), adapting to changes in daily routine (2304), managing one's time (2305), carrying out specified (d2308) and unspecified (d2309) daily routines. Routines require activities to doing in a temporal context involving order of activities, duration of each activity. While routines may become habits, it is not necessarily automatic. The study does not intend to study the relation among these, but to investigate instruments to assess time-related concepts ranging from basic orientation, concept of time to how a person is able to use these concepts in daily life as explained in literature (ICF and Janeslätt). We have modified this section of the introduction (pp. 4-5) for better clarity.

3. In the methods, I want to motivate the authors to work with the PICO criteria even when it is not straight-forward in this case. "P" and "C" is very clear. "I" would refer to the assessment (what types of assessment do you consider) and "O" would refer to the concept of time that is being measured. You could specify this under point 1. in the methods section. Especially because I think the authors will have to provide more details of the concept of time that the instruments measure in the data extraction (point 3.)

Response: The PICO is now incorporated under point 1, Table 1.

4. Regarding point 3. Data extraction, I suggest the authors to also extract the recruitment strategy, country, and the mode of the instrument (questionnaire, interview, observational ...).

Response: Country is already mentioned in 3a as Place (p. 11). Suggestions regarding and mode of the instrument are now incorporated in 3c under mode of administration.

5. In the discussion, what is the bigger goal? What exactly do you want to do with the tool in the future and how is it going to improve patients' lives?

Response: We are grateful for this valuable comment and we have added text on this in the discussion (p. 12).

Reviewer: 2

Dr. Deepak Nag Ayyala, Augusta University

Comments to the Author:

The aim of this protocol is to investigate various available on how time-related concepts are evaluated in understanding the time perception of older adults with cognitive impairments. The authors aim to perform a meta-analysis of studies published by screening various databases. The purpose of the study and the approach are very clearly presented. However, there are a few minor comments that can be addressed to improve the protocol.

Major comments:

1. For age: the selection criterion is not very stringent. The authors should define what “older adults” mean, (e.g. age greater than 50) use that for their search. Using average age might not be ideal and should be removed. If not possible, they should consider the median to be more robust to outliers.

Response: The definition of older adults varies from country to country and from study to study, with cutoffs such as 50 years and above and 65 years and above. Also, several studies may have both adults and older adults as participants. If the average age is high, the findings of such studies may be relevant even though a few younger people are included. Regarding the age criteria, we will therefore consider the results of the studies rather than the inclusion criteria. For these reasons, we have considered both options: ‘articles with older adults aged 60 years or older OR articles with average age of 60 years or older’. This is a selection criterion previously used by a national HTA organization in a review of social services for older adults (Dahlberg et al, 2019), now included as a reference in the manuscript (p. 7), and the cutoff of 60 years is based on WHO age criteria for defining older adults. By average, we intend to consider median or mean whichever is provided in the article.

Reference: Dahlberg, L., Ahlström, G., Bertilsson, G., & Fahlström, G. (2019). *Kunskapsläget för bedömning och insatser inom äldreomsorgen* [Social care and services for older adults. An evidence map addressing assessment and interventions]. Report 306/2019. Stockholm: Swedish Agency for Health Technology Assessment and Assessment of Social Services (SBU). Available at: <https://www.sbu.se/sv/publikationer/sbu-kartlagger/kunskapslaget-for-bedomning-och-insatser-inom-aldreomsorgen2/>

2. There should be a minimum sample size requirement for an article to be included in the analysis. Experimental protocols with a small number of replicates might introduce more variability into the analysis.

Response: The study not a meta-analysis. It is a scoping review and therefore does not require a minimum sample size requirement.

3. A detailed statistical analysis plan for doing the meta-analysis is not provided. The authors could provide some details on the overall plan of how to do the analysis before reporting the results.

Response: The study not a meta-analysis. It is a scoping review and data analysis is presented accordingly.

Minor comment:

1. On Page 10, line 50: the range of dates for the search is mentioned as September-October 2020. Please correct it to a future date if the data has not been collected yet. Otherwise, please provide preliminary data.

Response: We completed the search in September-October 2020. Screening is in progress but is unfortunately delayed due to unavoidable circumstances, including the Covid-19 pandemic. We are not able to provide preliminary data as of now. We have provided the search strategy for one database as supplementary material as suggested by the editor. We will do an updated search to address this delay and have mentioned this on page 8.

2. All the references are mentioned after a comma or a period, instead of before. Please fix them.

Response: This has now been corrected.

Reviewer: 1

Competing interests of Reviewer: None.

Reviewer: 2

Competing interests of Reviewer: N/A

VERSION 2 – REVIEW

REVIEWER	Ayyala, Deepak Nag Augusta University, Population Health Sciences
REVIEW RETURNED	30-Aug-2021
GENERAL COMMENTS	The authors have addressed all my concerns in their revision. I have no further comments.